# ProtoGNN: Prototype-Assisted Message Passing Framework for Non-Homophilous Graphs

## Abstract

Many well-known Graph Neural Network (GNN) models assume the underlying graphs are homophilous, where nodes share similar features and labels with their neighbours. They rely on message passing that iteratively aggregates neighbour's features and often suffer performance degradation on non-homophilous graphs where useful information is hardly available in the local neighbourhood. In addition, earlier studies show that in some cases, GNNs are even outperformed by Multi-Layer Perceptron, indicating insufficient exploitation of node feature information. Motivated by these two limitations, we propose ProtoGNN, a novel message passing framework that augments existing GNNs by effectively combining node features with structural information. ProtoGNN learns multiple class prototypes for each class from raw node features with the slot-attention mechanism. These prototype representations are then transferred onto the structural node features with explicit message passing to all non-training nodes irrespective of distance. This form of message passing, from training nodes to class prototypes to non-training nodes, also serves as a shortcut that bypasses local graph neighbourhoods and captures global information. ProtoGNN is a generic framework which can be applied onto any of the existing GNN backbones to improve node representations when node features are strong and local graph information is scarce. We demonstrate through extensive experiments that ProtoGNN brings performance improvement to various GNN backbones and achieves state-of-the-art performance on several non-homophilous datasets.

## 1 Introduction

Graph Neural Networks (GNNs) have emerged as prominent models for learning representations on graph-structured data. GNNs iteratively update the node embeddings based on the node features of its own and its local neighbors on the graph (Kipf & Welling, 2016; Defferrard et al., 2016; Gilmer et al., 2017). This form of iterative message passing provides strong structural inductive bias that assumes the presence of useful information in local neighborhoods. This assumption holds true for homophilous graphs whose connected nodes tend to share similar features and labels. However, many real-world graphs are structured in a heterophilous way wherein nodes tend to be dissimilar with their local neighbors. Such non-homophilous graphs can be found in domains of fraud detection (Shi et al., 2022; Pandit et al., 2007), molecular biology (Ye et al., 2022) and certain social networks (Lim et al., 2021).

Recent studies show that many GNNs (e.g. GCN (Kipf & Welling, 2016)) fail to learn well on non-homophilous graphs (Zhu et al., 2020b; 2021), and are even outperformed by simple architectures such as Multi-Layer Perceptron (MLP) that ignore graph structures and only leverage node features. This indicates two limitations of these GNNs. First, information of node features is not sufficiently exploited and might be diluted by iterative message passing. Second, the prevailing structural inductive bias that utilizes local neighbourhood information may not be helpful on non-homophilous graphs, since useful information is often scarce within the local neighbourhood.

Therefore, it is essential to go beyond the local neighborhood for useful information. Existing works either explicitly capture multi-hop information (Zhu et al., 2020b) or try to leverage globally available correlations with distant nodes (Suresh et al., 2021; Li et al., 2022). However, such methods often go beyond the linear complexity of GNN message passing (Suresh et al., 2021). More importantly,

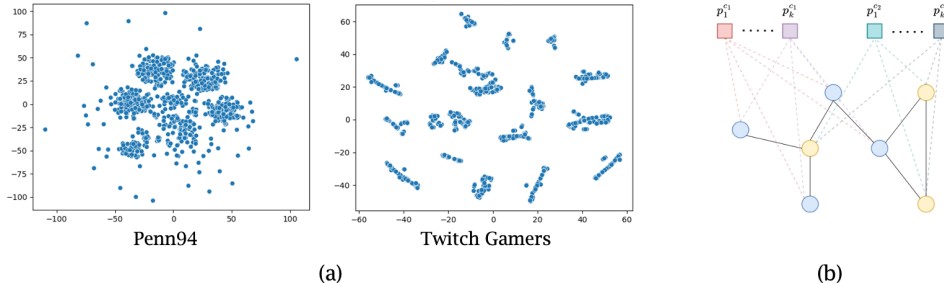

Figure 1: (a) Visualization of features for Penn94 and Twitch Gamers datasets. The plots show clustering patterns of the nodes in the feature space. (b) Illustration of information transfer from training nodes to prototypes. Yellow and blue represent different classes. The prototype nodes learn multiple representations per class from the training nodes and transfer the information to non-training nodes through attention.

since relevant information is often scattered within the vast amount of distant nodes, identifying and extracting such information is a challenging task.

To address the two limitations, we draw inspiration from prototypical networks (Snell et al., 2017), an efficient and effective paradigm that constructs a single representation for each class called a *prototype* to assist learning in data-scarce settings. Learning node representations on graph structures using class prototypes can facilitate information flow from distant nodes and capture global correlations efficiently. We observe that in many datasets, nodes from the same class form multiple clusters (see Fig. 1(a)). If we use a single prototype for each class, the prototype representation would be inaccurate or insufficient. For example, the single prototype may learn to represent the average of different clusters. Alternatively, the single prototype may learn to represent one of the dominant clusters in the feature space, therefore failing to capture information of the remaining clusters. Hence using a single prototype for each class may lead to suboptimal performance.

In this work, we propose ProtoGNN, a novel message-passing framework that augments existing GNNs and overcomes two limitations of GNNs on heterophilous graphs by facilitating efficient information transfer from distant nodes. In ProtoGNN, we first disentangle the two sources of information from the graph in the form of node feature view and structure view. With the node features of the training nodes, we construct multiple prototypes from each class independent of graph structure by adapting the mechanism of slot-attention (Locatello et al., 2020). These class prototypes are learnt from the node feature space via multiple rounds of attention. It provides an effective way of soft-clustering the node features as well as the needed model capacity for learning different clusters within the same class. Alternatively, this process can also be viewed as message passing from training nodes to prototype nodes as illustrated in Fig. 1(b). The learned prototype representations are then transferred onto all node representations from the structure view to make them more discriminative.

To enhance the effectiveness of ProtoGNN, we design two regularizers. To exploit the presence of the feature and structure views for regularization, we enforce cross-view compatibility between the prototypes (feature view) and node embeddings (structure view) through a hierarchical compatibility function which handles the presence of multiple prototypes within each class. To further exploit the full power of prototypes, we regularize them so that they are distinct from each other by encouraging the prototypes to be orthogonal to each other through an orthogonality loss.

The learning of prototypes introduces artificial edges from training nodes of each class to all non-training nodes via prototype nodes. This serves as a shortcut in message passing which bypasses local graph neighbourhoods and captures global information, while maintaining the linear complexity with respect to the backbone GNNs. Additionally, since the prototypes are learnt from node feature space, it preserves strong feature information that might be diluted by traditional message passing.

Overall, ProtoGNN is a generic framework which presents the following advantages: (1) It is orthogonal to existing GNN backbones and can be applied onto any of them to improve node representations in heterophilous graphs. (2) It is efficient and can capture global node correlations with only $\mathcal{O}(n)$ additional edges. (3) It can be useful in graphs where distant information is helpful and node feature information is strong, even in homophilous settings when label rate is low. (4) It preserves the node features from dilution due to message passing. We conduct extensive experiments

on several non-homophilous datasets which show consistent improvement over various backbone GNN models, achieving state-of-the-art performance in multiple instances.

## 2 RELATED WORKS

GNN models like GCN (Kipf & Welling, 2016) and GAT (Veličković et al., 2017) have performed profoundly well on graph-structured data in a multitude of domains. However, GCN-like models have an intrinsic assumption of homophily on graphs, and it has been noted by multiple works that they perform poorly on heterophilous graphs (Pei et al., 2020; Zhu et al., 2020b). Early works tried to address this issue either through multi-hop neighbourhood aggregation or building auxiliary graph structures based on node and structural features. Since then, various ways have been proposed to make GNNs work better on non-homophilous graphs (Bo et al., 2021; Yan et al., 2021; Chien et al., 2020; Dong et al., 2021; Suresh et al., 2021; Yang et al., 2021; Lim et al., 2021; Luan et al., 2021; Zhu et al., 2020a; Liu et al., 2021). Some of these methods leverage global information by letting the model choose either proximity or structural information or both for prediction (Suresh et al., 2021), while other methods like GGCN (Yan et al., 2021) address this by introducing signed messages in the propagation scheme based on over-smoothing effect observed in GCN-like models.

More recently, LINKX (Zhu et al., 2021) showed that a simple MLP applied on node features and adjacency matrix can perform well on many heterophilous graph datasets. In addition, GloGNN (Li et al., 2022) showed substantial improvements when further global information is captured. However, despite introducing an efficient approach to do this, it can take quadratic runtime when the graph is dense. In this work, we capture relevant information from distant nodes, which is often scattered within the graph structure. Identifying and extracting such information is a challenging task. Inspired by prototypical networks (Snell et al., 2017), we construct class prototypes to enable efficient information transfer from distant nodes with linear complexity. Furthermore, with the observation that node features occur in several clusters in heterophilous graphs, we learn multiple prototypes for each class with an adapted form of slot-attention (Locatello et al., 2020). There are also a few works that use prototype-based learning in graphs. One uses prototypes to make the predictions more explainable (Zhang et al., 2022) while another considers few-shot setting (Ding et al., 2020). However, they do not integrate prototype learning with backbone GNNs, use multiple prototypes per class or consider heterophilous graphs. Very recently, Xu et al. (2022) proposed using memory units to capture global patterns in the graph. However, unlike Xu et al. (2022), we train multiple prototypes per class and are inductively-biased with slot attention mechanism to capture node clusters.

## 3 PRELIMINARIES

**Notations.** Let $\mathcal{G} = (\mathcal{V}, \mathcal{E})$ represent an undirected graph, where $\mathcal{V} = \{v_i\}_{i=1}^n$ is the set of nodes and $\mathcal{E} \subseteq \mathcal{V} \times \mathcal{V}$ is the set of edges on the graph. $A$ denotes the adjacency matrix of the graph, where $A_{ij}$ represents the edge $e_{ij}$ between nodes $v_i$ and $v_j$. $X$ represents the feature matrix of nodes, where $X_i$ is the features of node $i$, and $X_{train}$ and $X_{nontrain}$ are the feature matrices for all the training and non-training nodes, respectively. $Y$ denotes the node class labels, and $y_i$ is the label for node $v_i$.

**Homophily and heterophily** Homophily refers to the degree of similarity between connected neighboring nodes in terms of their features or labels. There are many types of homophily measures proposed, including edge homophily (Zhu et al., 2020b), node homophily (Pei et al., 2020), and class insensitive homophily (Lim et al., 2021). In this work, we use the homophily matrix proposed in Lim et al. (2021), since it can better reflect class-wise homophily. The homophiliy matrix is defined as:

$$H_{c_1,c_2} = \frac{|(u,v) \in E : c_u = c_1,\ c_v = c_2|}{|(u,v) \in E : c_u = c_1|},$$

(1)

for classes $c_1$ and $c_2$, $H_{c_1 c_2}$ denotes the proportion of edges between from nodes of class $c_1$ to nodes of class $c_2$. A homophilous graph has high values on the diagonal entries.

## 4 PROTOGNN

With the aim of capturing scattered but useful information from distant nodes without diluting the strong node features, we propose ProtoGNN, which learns the node representations by decoupling

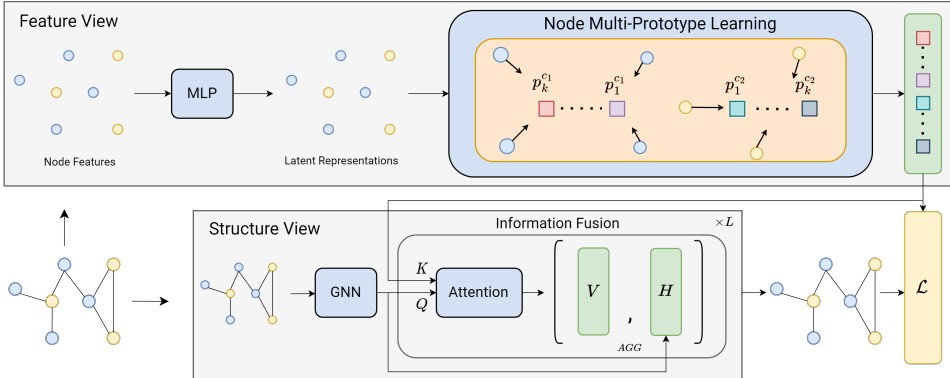

Figure 2: **Overall architecture of ProtoGNN:** ProtoGNN disentangles the input graph into two views, one considering only node features i.e. *feature view*, the other considering both graph structure and node features i.e. the *structure view*. In the feature view, ProtoGNN learns multiple prototype representations for each class from the node features. In the structure view, a backbone GNN generates node embeddings with the assistance of information from the feature view. After each GNN layer, generated node embeddings could be augmented by attending to prototypes. The whole pipeline is trained with a combination of three loss functions.

the feature view and the structural view from the graph and fusing them to produce effective representations. ProtoGNN can be broken down into the following key components: (1) a feature view which involves learning multiple prototype representations for each class from only node features, (2) a structure view that consists of *any* GNN model, (3) information fusion which combines information from the two views via the attention mechanism. The whole pipeline is trained with three loss functions. In the following, we discuss the three stages in details.

**Feature View:** In order to fully exploit the node features and be able to transfer information to distant nodes, we learn prototype representations of each class without the structural inductive bias. This forms the feature view of the graph, wherein nodes are clustered in feature space irrespective of their topological affinities. The prototype features can then be used to transfer information onto the structural representations for the model to learn the most effective embeddings adaptively.

In real-world graphs, we observe that the feature distribution for each class of nodes is usually multi-modal, as illustrated by the feature space visualization in Fig. 1(a). Using a single prototype to represent multi-modal distributions is insufficient and often inaccurate. Therefore, we design multiple prototype nodes for each class to enable them to capture different modes in feature distribution. The embeddings of these prototype nodes are learned via an adapted slot attention mechanism (Locatello et al., 2020). In the original setting, a fixed number of slots are learnt to represent different objects in computer vision tasks.

In ProtoGNN, we extend slot attention mechanism to learn the prototypes/clusters of node features of each class. The learning is done via attention where the class prototypes compete for explanations of each node, resulting in soft clustering effects such that each slot dominates in certain groups of nodes.

For learning the prototypes, we first transform the node features using an MLP to generate the node embeddings. These embeddings are averaged and used to initialize the mean $\boldsymbol{\mu}$ of a Gaussian distribution $\mathcal{N}(\boldsymbol{\mu}, \boldsymbol{\Sigma})$ with randomly initialized and learned co-variance $\boldsymbol{\Sigma}$. We initialize $K$ prototypes for each class $c$, denoted by $P^c \in \mathbb{R}^{K \times d}$, by sampling from the distribution $\mathcal{N}(\boldsymbol{\mu}, \boldsymbol{\Sigma})$. We then iteratively update the embeddings. In each iteration, we compute a dot product attention (Luong et al., 2015) between the prototypes and nodes, followed by a softmax operation that normalizes over the prototypes, such that the assignment scores to each prototype sum up to one for a given node. In this way, the prototypes will compete for an assignment of nodes which acts as a soft clustering on the node feature space. Specifically, the attention weights $M_{attn}^c$ for class $c$ is

$$X = MLP(X_{raw}), \tag{2}$$

$$M_{attn}^c = \text{Softmax}\left(\frac{[P^c W^Q] \cdot [W^K (X_{train}^c)^T]}{\sqrt{d}}\right) + \epsilon \quad \in \mathbb{R}^{K \times N_{train}^c}, \tag{3}$$

where $d$ is the hidden dimension, $N_{train}^c$ is the number of training nodes belonging to class $c$, $W^Q$ and $W^K$ are query and key transformation matrices respectively, and $\epsilon$ is a small coefficient added for numerical stability. Then, we normalize the attention matrix $M_{attn}^c$ over the training node dimension, so that for each prototype, its assigned weights in all training nodes sum up to one. We then update the prototype embeddings $P^c$ using the weighted average of training node features $X_{train}^c$.

$$P^c = \text{Norm}(M_{attn}^c) X_{train}^c W^V \quad \in \mathbb{R}^{K \times d} \tag{4}$$

where $W^V$ is a parameter matrix. The learning of prototypes serves as a soft clustering of feature space. It can also be viewed as message passing from training nodes to prototypes via artificially added edges.

**Structure View.** We use GNN to learn the structure view representation of nodes. GNN takes both node features and adjacency matrix as inputs. After each layer of message passing or processing, GNN produces a structure view node embedding matrix $H^l \in \mathbb{R}^{N \times d}$. Note that we can use any GNN as a backbone to generate the structure view embeddings.

$$H^l = GNN^l(A, H^{l-1}) \qquad \text{where} \quad H^0 = X_{raw} \tag{5}$$

**Information Transfer/Fusion.** Once we have all the class prototypes $P$ from the feature view and node embeddings $H$ from the structure view, we can then transfer the information from prototype nodes onto the structure view embeddings. This is done by first computing dot product attention between prototypes from all classes and all non-training nodes, and then using the softmax function to normalize over the prototype dimension, so that the weight of all prototypes sum up to one for each non-training node.

$$S_{attn} = \text{Softmax}(H_{nontrain}^l P^T) \quad \in \mathbb{R}^{N_{nontrain} \times CK} \tag{6}$$

Next, we compute the message to be passed from prototype nodes to non-training nodes by taking the weighted average of prototype embeddings according to the dot product attention weights. Finally, we fuse the prototype messages from the feature view with the node embeddings from the structure view through an aggregation function. We allow the structure view of node embeddings to be taken from any layer of the GNN, making it an architectural choice.

$$H^l = \text{AGG}(H^l, M_{proto}) \qquad \text{where} \quad M_{proto} = S_{attn} P \tag{7}$$

This information transfer process can also be viewed as a message passing from prototype nodes to non-training nodes. Therefore, the ProtoGNN framework essentially constructs a shortcut between all the nodes in the graph. This is especially beneficial in the non-homophilous setting where useful information is less available in the graph local neighborhood. This also helps in some homophilous settings as well when useful information is distant.

**Training.** To exploit the full potential of the multiple prototypes of each class, we regularize training with two additional loss functions besides the cross-entropy loss. First, the cross-view *compatibility loss* regularizes by encouraging the structure and feature views to agree. Second, the *orthogonality loss* encourages the prototypes to be distinct from each other.

We modify the compatibility loss function from Snell et al. (2017), making it hierarchical in order to handle multiple prototypes within each class. This is done with the help of an additional aggregation function. We first compute the similarities between the prototypes of a class and a node embedding in order to compute the likelihood that the embedding belongs to the class. The similarity scores with the multiple prototypes are then aggregated into a single score for the class. Multiple choices of aggregators are possible, e.g. we could compute the average similarities between the embedding and

the prototypes in the class. We use the max aggregator in order to *select* the most similar prototype within the class to the node. The compatibility loss is then computed as follows:

$$s_i^c = \max_{k \in K} \Big( \Lambda(f(h_i^L), p_k^c) \Big) \tag{8}$$

$$\mathcal{L}_{comp} = \frac{1}{CN_{train}} \sum_{i \in N} \left[ s_i^{c_i} + \log \sum_{c' \neq c} \exp\big( - s_i^{c'} \big) \right] \tag{9}$$

where $N_{train}$ is the number of training nodes, $C$ is number of classes, $c_i$ is the class-label of $i^{th}$ training node, $p_k^c$ is the $k^{th}$ prototype of class $c$, $h_i^L$ is the node embedding before final classification, $f$ is an MLP and $\Lambda$ is a similarity function for which we use dot product. To exploit the full power of prototypes, we regularize them so that they are distinct from each other. We do that by enforcing the prototypes to be orthogonal to each other through an orthogonality loss shown to be useful according to Bianchi et al. (2020).

$$\mathcal{L}_{ortho} = \left\| \frac{P^{c^T} P^c}{\|P^{c^T} P^c\|_F} - \frac{I_d}{\sqrt{d}} \right\|_F \tag{10}$$

where $\|\cdot\|_F$ indicates the Frobenius norm. With the $\alpha$ and $\beta$ as tuning hyperparameters, the full loss function used for training is

$$\mathcal{L} = \mathcal{L}_{ce} + \alpha \mathcal{L}_{comp} + \beta \mathcal{L}_{ortho} \tag{11}$$

**Complexity.** In Tab. 1, we compare the complexity of ProtoGNN with other baseline models which add additional edges. Particularly, GloGNN (Li et al., 2022), though efficient in sparse graphs with small $d$ (average degree), can go quadratic for dense graphs.

Table 1: comparison

| | H2GCN | WR-GAT | GLoGNN | PROTOGNN |
|---|---|---|---|---|
| MEAN | $\mathcal{O}(n^2)$ | $\mathcal{O}(n \log n)$ | $\mathcal{O}(dCn)$ | $\mathcal{O}(KCn)$ |

ProtoGNN captures global information with only $\mathcal{O}(n)$ additional edges. More specifically, it contains additional $KCn$ virtual edges, when the graph contains $C$ classes with $K$ prototypes per class. Note that both $K, C \ll n$ and hence, it preserves the linear complexity of GNNs.

## 5 EXPERIMENTS

### 5.1 EXPERIMENT SETUP

**Datasets** We conduct experiments on 10 datasets, including 7 non-homophilous datasets: Penn94, Pokec, Genius, Twitch-gamers, Cornell5 and Amherst41 released by Lim et al. (2021), and US-election dataset that we identify to be non-homophilous (Jia & Benson, 2020). We use the original train/validation/test splits if such splits exist. The train/validation/test split is 60%, 20%, 20% for US-election, and 50%, 25%, 25% for other datasets. We also evaluate the performance of ProtoGNN on three popular homophilous datasets: Cora, Citeseer and Pubmed (Pei et al., 2020), with a low label rate setting so that a labeled node is unlikely to be close to a node that needs to be labeled. The statistics of the datasets are included in Tab. 2. More details of the datasets are in the appendix.

**Baselines** ProtoGNN is a generic framework that can be applied onto any of the existing GNN backbones. We first add ProtoGNN to 4 backbones: (1) GCN (Kipf & Welling, 2016); (2) GCN with batch normalization, as we observe that batch normalization significantly improves performance in some datasets; (3) GCNII (Chen et al., 2020), a deeper GNN architecture; (4) LINKX (Lim et al., 2021), a model specifically designed for non-homophilous graphs. Following Lim et al. (2021) and Li et al. (2022), we also compare ProtoGNN against 11 other baselines for comprehensive evaluation, including (1) MLP; (2) general GNNs, including GCN, GAT (Veličković et al., 2017), MixHop (Abu-El-Haija et al., 2019) and GCNII; (3) baselines designed for non-homophilous graphs, including H$_2$GCN (Zhu et al., 2020b), WRGAT (Suresh et al., 2021), GPR-GNN (Chien et al., 2020) ACM-GCN (Luan et al., 2021), LINKX and GloGNN (Li et al., 2022).

Table 2: Statistics of non-homophilous datasets and Classification performance of ProtoGNN vs backbones on non-homophilous datasets. Hom. Mat. refers to the homophily matrix described in Sec. 3. 0 and 1 stand for class labels. We report AUC for Genius (Lim et al., 2021) and accuracy for other datasets with standard deviation over 3 trials. OOM refers to the out-of-memory error.

| | Penn94 | Pokec | Genius | Twitch-gamers | Cornell5 | Amherst41 | US-election |
|---|---|---|---|---|---|---|---|
| **#Nodes** | 41,554 | 1,632,803 | 421,961 | 168,114 | 18660 | 2235 | 3234 |
| **#Edges** | 1,362,229 | 30,622,564 | 984,979 | 6,797,557 | 790,777 | 90,954 | 11,100 |
| **Hom. Mat.** (0,0 / 0,1) | 0.52 / 0.48 | 0.50 / 0.50 | 0.88 / 0.12 | 0.51 / 0.50 | 0.57 / 0.43 | 0.55 / 0.45 | 0.59 / 0.41 |
| **Hom. Mat.** (1,0 / 1,1) | 0.50 / 0.50 | 0.61 / 0.39 | 0.85 / 0.15 | 0.41 / 0.59 | 0.48 / 0.52 | 0.49 / 0.51 | 0.12 / 0.88 |
| GCN | 77.08 (0.40) | 66.52 (5.16) | **87.09** (0.51) | 61.78 (0.12) | 73.91 (0.32) | **81.10** (1.51) | 81.71 (2.55) |
| +ProtoGNN | **81.43** (0.26) | **73.59** (0.30) | 86.78 (1.16) | **62.53** (0.32) | **80.19** (0.17) | 81.03 (0.92) | **83.82** (1.34) |
| $GCN_{BN}$ | 82.47 (0.27) | 75.45 (0.17) | **87.42** (0.37) | 62.18 (0.26) | 80.15 (0.37) | 81.41 (1.70) | 82.07 (1.65) |
| +ProtoGNN | **82.97** (0.07) | **76.91** (0.03) | 87.16 (0.94) | **62.62** (0.16) | **80.73** (0.48) | **81.56** (2.00) | **82.74** (1.35) |
| GCNII | 82.92 (0.59) | OOM | 90.24 (0.29) | 63.39 (0.61) | 78.85 (0.78) | 76.02 (1.38) | 82.90 (0.29) |
| +ProtoGNN | **83.19** (0.31) | OOM | **91.18** (0.13) | **65.20** (0.68) | **80.08** (0.48) | **79.13** (0.74) | **84.08** (0.46) |
| LINKX | 84.71 (0.52) | **82.04** (0.07) | 90.77 (0.27) | 66.06 (0.19) | 83.42 (0.59) | 81.57 (1.04) | 84.08 (0.67) |
| +ProtoGNN | **86.23** (0.34) | 81.94 (0.10) | **90.80** (0.10) | **66.42** (0.34) | **84.14** (0.13) | **82.81** (1.34) | **86.97** (1.21) |

## 5.2 EXPERIMENTAL RESULTS

### 5.2.1 USING PROTOGNN TO ENHANCE EXISTING GNN BACKBONES

We are motivated to design ProtoGNN as a generic framework that is orthogonal to existing GNN models to boost their performance on heterophilous graphs. We therefore evaluate the effects of applying ProtoGNN onto various existing GNN backbones. We first apply ProtoGNN on GCN, one of the most widely used general-purpose GNNs, and observe performance improvement on 5 out of 7 datasets, with substantial gains on Penn94, Pokec, Cornell5, and US-election (Tab. 2). During experiments on GCN, we find that batch normalisation brings consistent performance boost to GCN on various non-homophilous datasets. We therefore add ProtoGNN on top of GCN with batch normalization to evaluate its performance on this GCN variant. $GCN_{BN}$+ProtoGNN achieves consistent performance improvement on all datasets except Genius.

We then apply ProtoGNN on GCNII, a deeper GNN model that is capable of retrieving global information by propagation through the additional layers. Though not designed for heterophilous graphs, it serves as strong baseline for heterophilous datasets. GCNII+ProtoGNN achieves better performance than the original GCNII on all datasets, with pronounced improvement on Amherst41.

We next evaluate our framework's effectiveness on a backbone explicitly designed for heterophilous graphs. LINKX+ProtoGNN performs consistently better than LINKX on 6 out of 7 datasets, with substantial enhancement on US-election. Penn94 and Amherst41 datasets. Overall, the results demonstrate consistent improvement of ProtoGNN on various backbone GNNs and datasets.

### 5.2.2 COMPARISON AGAINST MORE BASELINES

Table 3: Classification performance comparison against baselines. Notations follow style in Tab.2.

| | Penn94 | Pokec | Genius | Twitch-gamers | Cornell5 | Amherst41 | US-election |
|---|---|---|---|---|---|---|---|
| MLP | 73.61 (0.40) | 62.37 (0.02) | 86.68 (0.09) | 60.92 (0.07) | 68.86 (1.83) | 60.43 (1.26) | 81.92 (1.01) |
| GCN | 82.47 (0.27) | 75.45 (0.17) | 87.42 (0.37) | 62.18 (0.26) | 80.15 (0.37) | 81.41 (1.70) | 82.07 (1.65) |
| GAT | 81.53 (0.55) | 71.77 (6.18) | 55.80 (0.87) | 59.89 (4.12) | 78.96 (1.57) | 79.33 (2.09) | 84.17 (0.98) |
| MixHop | 83.47 (0.71) | 81.07 (0.16) | 90.58 (0.16) | 65.64 (0.27) | 78.52 (1.22) | 76.26 (2.56) | 85.90 (1.55) |
| GCNII | 82.92 (0.59) | OOM | 90.24 (0.08) | 63.39 (0.61) | 78.85 (0.78) | 76.02 (1.38) | 82.90 (0.29) |
| H²GCN | 81.31 (0.60) | OOM | OOM | OOM | 78.46 (0.75) | 79.64 (1.63) | 85.53 (0.77) |
| WRGAT | 74.32 (0.53) | OOM | OOM | OOM | 71.11 (0.48) | 62.59 (2.46) | 84.45 (0.56) |
| GPR-GNN | 81.38 (0.16) | 78.83 (0.05) | 90.05 (0.31) | 61.89 (0.29) | 73.30 (1.87) | 67.00 (1.92) | 84.49 (1.09) |
| ACM-GCN | 82.52 (0.96) | 63.81 (5.20) | 80.33 (3.91) | 62.01 (0.73) | 78.17 (1.42) | 70.11 (2.10) | 85.14 (1.33) |
| LINKX | 84.71 (0.52) | 82.04 (0.07) | 90.77 (0.27) | 66.06 (0.19) | 83.46 (0.61) | 81.73 (1.94) | 84.08 (0.67) |
| GloGNN++ | 85.74 (0.42) | **83.05** (0.07) | 90.91 (0.13) | 66.34 (0.29) | 83.96 (0.46) | 81.81 (1.50) | 85.48 (1.19) |
| ProtoGNN | **86.23** (0.34) | 81.94 (0.10) | **91.18** (0.13) | **66.42** (0.34) | **84.14** (0.13) | **82.81** (1.34) | **86.97** (1.21) |

To further evaluate the effectiveness of ProtoGNN, we compare our best results in Tab. 2 against 11 strong baselines following Lim et al. (2021) and Li et al. (2022), including MLP and 4 general-purpose GNNs: GCN, GAT, MixHop and GCNII. The remaining 6 baselines are GNNs built

specifically for heterophilous graphs, including $H_2$GCN, WRGAT, GPR-GNN, ACM-GCN, LinkX and GloGNN++. All ProtoGNN results in Tab. 3 are from LINKX+ProtoGNN, except for Genius where GCNII+ProtoGNN performs best.

ProtoGNN achieves state-of-the-art on 6 out of 7 datasets compared to all baselines, except on Pokec. While some methods incur OOM errors when running on large-scale datasets, ProtoGNN does not suffer OOM due to its linear complexity with respect to backbones.

### 5.2.3 IMPACT OF SHORTCUT CONSTRUCTION

An important property of ProtoGNN is introducing shortcuts between training nodes and non-training nodes via the constructed class prototypes. By facilitating the transfer of useful information from afar, shortcuts become especially helpful on heterophilous graphs, where useful information is often unavailable in local neighbourhoods. We use the US-election dataset to illustrate the effect of shortcuts. Here, nodes represent counties, and edges connect bordering counties. Red and blue colors represent the election outcomes of the counties. As visualized in Fig. 3, nodes on the east coast and west coast are separated by a swathe of red counties, but share similar node features

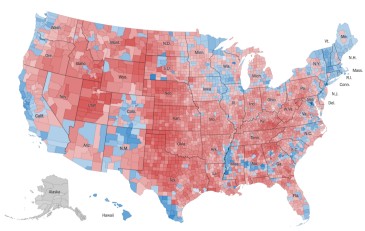

Figure 3: US 2012 election results map by county.

such as income, education and demographics. A shortcut between them via prototypes helps the message bypass the red counties in middle America, preventing message dilution in traditional message passing neural networks. Our results in Tab. 2 show that ProtoGNN achieves consistent performance improvement on US-election when added to all GNN backbones.

Table 4: Classification accuracy on Cora, Citeseer and Pubmed with 5 labels per class.

|  | CORA | CITESEER | PUBMED |
|---|---|---|---|
| GCN | 69.23 (3.39) | 63.03 (4.48) | 68.00 (3.75) |
| +PROTOGNN | **71.79** (1.56) | **64.00** (3.45) | **72.67** (3.88) |

To further evaluate the effects of shortcut construction, we consider a generalized scenario on homophilous graphs where information from labeled nodes needs to propagate a long distance to reach most unlabeled nodes. Specifically, we evaluate our method on Cora, CiteSeer and Pubmed with only a small number of training labels per class. We use a single prototype for each class without slot attention and obtain prototype representations by taking the average of training node features, since the training labels are scarce. Our hypothesis is that information propagation becomes more challenging with increasing scarcity of useful training information, making the effects of shortcut construction more pronounced in sparsely-annotated graph datasets. The hypothesis is confirmed by experiment results. As shown in Tab. 4, GCN+ProtoGNN beats vanilla GCN on all 3 datasets, with substantial improvements on Cora and Pubmed.

### 5.2.4 ABLATION STUDIES

**Number of prototypes per class**    Motivated by the observation that node features tend to form clusters in feature space illustrated in Fig. 1, we assign multiple prototypes per class to capture the clusters. Our hypothesis is that multi-modal feature distribution is more accurately represented using multiple prototypes. We test the hypothesis by varying the number of prototypes per class in ablative experiments. We train ProtoGNN with combinations of different numbers of prototypes and hidden dimensions. Fig. 4 illustrates ablative experiments of LINKX+ProtoGNN on Penn94. We observe that: (1) ProtoGNN with a single prototype per class outperforms the backbone. (2) ProtoGNN with multiple prototypes per class perform consistently better than their single prototype counterparts. Our

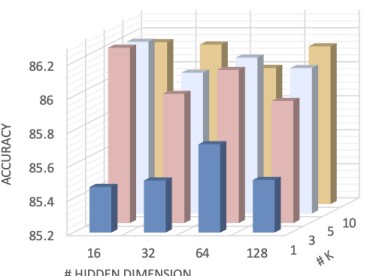

Figure 4: Effects of the number of prototypes per class(K).

ablative experiments shows the effectiveness of using prototypes, as well as the benefits of using multiple prototypes per class over using a single one. (3) The optimal number of slots depends on a number of factors, such as hidden dimension, dataset feature space and backbone model type.

**Loss functions** To achieve the full potential of the soft-clustering objective of our model, we add two additional losses to learn prototypes effectively. The *compatibility loss* enforces similarity in embeddings, while the *orthogonality loss* acts as a regularizer constraining the model to learn distinct prototype clusters for each class. In this ablation study, we evaluate the effect of these losses on model performance. We vary

Table 5: Effects of Compatibility and Orthogonality losses on Amherst41 and Cornell5 datasets.

| $\mathcal{L}_{comp}$ $(\alpha)$ | $\mathcal{L}_{ortho}$ $(\beta)$ | **AMHERST41** | **CORNELL5** |
|---|---|---|---|
| 0 | 0 | 81.36 (0.09) | 83.05 (0.96) |
| 0.1 | 0.01 | 81.88 (0.32) | 84.14 (0.13) |
| 0.01 | 0.1 | 82.28 (0.73) | 83.89 (0.12) |
| 0.01 | 0.01 | 82.28 (0.42) | 83.76 (0.16) |

$\alpha$ and $\beta$ parameters in Eqn. 11 from the set $\{0, 0.01, 0.1\}$, with hidden dimension fixed to $64$ and the number of slots set to $5$. Results in Tab. 5 show consistent performance improvements for all non-zero values of $\alpha$ and $\beta$ on both datasets, compared to cross-entropy loss alone. This provides substantial evidence for the usefulness of the additional loss functions.

**Aggregation operation in Compatibility loss** In order to make the node embeddings similar to prototype representations from the same class, we design the cross-view *compatibility loss $\mathcal{L}_{comp}$*. After computing the similarity between a node and each of the multiple prototypes from the same class, the choice of aggregation method is critical. We evaluate the effectiveness of using the aggregation operator on Amherst41 and Cornell5 datasets. Tab. 6 shows the effects of replacing the max aggregator with mean

Table 6: Effects of aggregator function used in Compatibility loss on Amherst41 and Cornell5 datasets.

| $AGG$ | **AMHERST41** | **CORNELL5** |
|---|---|---|
| MEAN | 81.17 (0.91) | 82.73 (0.35) |
| SUM | 81.17 (1.73) | 83.53 (0.36) |
| MAX | 82.81 (1.34) | 84.14 (0.13) |

and sum in computing compatibility loss. On both datasets, max outperforms both sum and mean, indicating the effectiveness of using max as the aggregation operation. Intuitively, taking the average of all similarity scores (mean) is sub-optimal. mean tends to make the node embeddings closer to the average of all prototypes from the same class, undermining the purpose of using multiple prototypes to represent different clusters. Similar to mean, summing up all similarity scores (sum) is more powerful yet requires more data to learn. max selects the maximum similarity score to compute compatibility loss and guides the node embeddings closer to one of the class prototypes, thus preserving the power of diversity in representation.

**Node multi-prototype learning.** We adapt slot-attention mechanism to learn multiple prototypes for each class. In order to evaluate the effectiveness of learning prototypes with slot-attention, we ablate the model against two variants: 1)ProtoGNN-Naive, a variant where class prototypes are constructed by taking the average of all training node features from that

Table 7: Effect of Slot Attention on Amherst41 and Cornell5 datasets.

| SETTINGS | **AMHERST41** | **CORNELL5** |
|---|---|---|
| PROTOGNN-NAIVE | 80.84 (1.94) | 83.94 (0.08) |
| PROTOGNN-SA | 82.15 (1.07) | 83.25 (0.14) |
| PROTOGNN | 82.81 (1.34) | 84.14 (0.13) |

class. 2) ProtoGNN-SA, where we adopt the classic self attention mechanism to learn class prototypes. We show results on Amherst41 and Cornell5 datasets keeping other hyperparameters fixed in both models. As shown in Tab. 7, we observe that ProtoGNN consistently outperforms both variants, signifying the effectiveness of using slot attention. Its better performance may be attributed to the ability to perform soft clustering that enables the multiple prototypes per class to capture the underlying clusters in feature space, which self-attention and taking the average of node features cannot do. A detailed comparison between ProtoGNN-SA and ProtoGNN can be found in Appendix.

## 6 CONCLUSION

In this work, we address the two challenges in heterophilic graphs of efficiently capturing scattered but useful global information from distant nodes and preserving node feature information from dilution. For this, we propose ProtoGNN, a prototype-assisted message passing framework that augments existing GNNs by effectively combining node feature information with structural information. ProtoGNN exploits strong node features by constructing multiple class prototypes from training nodes. It then passes the information from prototype nodes to non-training nodes, updating their structural embeddings learned through GNN backbones. It thus creates shortcuts that bypass the local graph neighbourhoods and efficiently captures global information. ProtoGNN is a generic framework that can be applied to any existing GNN backbones. Our extensive experiments demonstrate that ProtoGNN brings consistent improvements when applied to various GNN backbones, achieving state-of-the-art results on multiple datasets against strong baselines.

ETHICS STATEMENT

Our work is a generic framework to improve node representation learning on non-homophilous graphs. This work applies to GNNs generally and non-homophilous graph structured data specifically. Its ethical impacts (including positive and negative) depends on the specific domain of the data. We conduct our experiments on publicly available datasets in compliance with fair-use clauses.

REPRODUCIBILITY STATEMENT

Details of the datasets used can be found in Appendix A.1. For experiments, the settings can be found in Section 5.1 and details of the hyperparameter settings used for training are specified in Appendix A.2. To further facilitate reproducibility of this work, we will make the code publicly available at anonymised repository https://github.com/protognn/protognn.

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

## A APPENDIX

### A.1 DATASET DETAILS

Penn94, Cornell5 and Amherst41 (Lim et al., 2021) are friendship network datasets extracted from Facebook of students from selected universities from 2005. Each node in the datasets represent a student, while node label represents the reported gender of the student. Node features include major, second major/minor, dorm/house, year, and high school.

Pokec (Lim et al., 2021) is a friendship graph dataset from a popular Slovak online social network, where nodes represent users and edges represent directed friendship relations. Node label represents the reported gender of user. Node features include geographical region, registration time, and age.

Genius (Lim et al., 2021) is a sub-network from website genius.com, a crowd-sourced website of song lyrics annotations. Nodes represent users while edges connect users that follow each other. Node features include expertise scores expertise scores, counts of contributions and roles held by users. Around 20% of the users are marked with a "gone" label, indicating that they are more likely to be spam users. The task is to predict which users are marked.

Twitch-gamers (Lim et al., 2021) is a sub-graph extracted from the streaming platform Twitch. Each node represents a Twitch account, while edges connect accounts that are mutual followers. Node features include number of views, creation and update dates, language, life time, and whether account is dead. The task is to predict whether the account has explicit content.

US-election (Jia & Benson, 2020) is a geographical dataset extracted from statistics of Unite States election of year 2012. Nodes represent US counties, while edges connect bordering counties. Node features include income, education, population etc. The task is a binary classification to predict election outcome.

Cora, Citeseer and Pubmed (Pei et al., 2020) are citation graphs, where each node represents a scientific paper and two papers are connected when a paper cites the other. Each node is labeled with the research field and the task is to predict which field the paper belongs to. All three datasets are homophilous.

### A.2 HYPERPARAMETE DETAILS

- MLP: hidden dimension $\in \{16, 32, 64\}$, number of layers $\in \{2, 3\}$. Activation function is ReLU.
- GCN: lr $\in \{.01, .001\}$, hidden dimension $\in \{4, 8, 16, 32, 64\}$, except for Pokec, where hidden dimension $\in \{4, 8, 16\}$. number of layers $\in \{2\}$. Activation function is ReLU.
- GAT: lr $\in \{.01, .001\}$. For Pokec: hidden channels $\in \{4, 8, 12\}$ and gat heads $\in \{2, 4\}$. For the remaining datasets: hidden channels $\in \{4, 8, 12, 32\}$ and gat heads $\in \{2, 4, 8\}$. number of layers $\in \{2\}$. We use the ELU as activation.
- MixHop: hidden dimension $\in \{8, 16, 32\}$, number of layers $\in \{2\}$.
- GCNII: number of layers $\in \{2, 8, 16, 32, 64\}$, strength of initial residual connection $\alpha \in \{0.1, 0.2, 0.5\}$, hyperparameter for strength of the identity mapping $\theta \in \{0.5, 1.0, 1.5\}$.
- H$_2$GCN: hidden dimension $\in \{16, 32\}$, dropout $\in \{0, .5\}$, number of layers $\in \{1, 2\}$. Model architecture follows Section 3.2 of Zhu et al. (2020b).
- WRGAT: lr $\in \{.01\}$, hidden dimension $\in \{32\}$.
- GPR-GNN: lr $\in \{.01, .05, .002\}$, hidden dimension $\in \{16, 32\}$.
- ACM-GCN: lr $\in \{.01\}$, weight decay $\in \{5e^{-5}, 5e^{-4}, 5e^{-3}\}$, dropout $\in \{0.1, 0.3, 0.5, 0.7, 0.9\}$, hidden channels $\in \{64\}$, number of layers $\in \{2\}$, display step $\in \{1\}$.
- LINKX: hidden dimension $\in \{16, 32, 64\}$, number of layers $\in \{1, 2\}$. Rest of the hyperparameter settings follow Lim et al. (2021).
- GloGNN++: lr $\in \{.001, .005, .01\}$, weight decay $\in \{0, .01, .1\}$, dropout $\in \{0, .5, .8\}$, hidden channels $\in \{128, 256\}$, number of layers $\in \{1, 2\}$, $\alpha \in \{0, 1\}$, $\beta \in \{0.1, 1\}$,

$\gamma \in \{0.2, 0.5, 0.9\}$, $\delta \in \{0.2, 0.5\}$, number of normalization layers $\in \{1, 2\}$, orders $\in \{1, 2, 3\}$, display step $\in \{1\}$.

- ProtoGNN: lr $\in \{.005, .01\}$, weight decay $\in \{0, 0.001, .0005\}$, loss coefficients $\alpha, \beta \in \{0, 0.01, 0.1\}$, prototypes per class $(K) \in \{1, 3, 5, 10\}$, hidden dimension $\in \{16, 32, 64\}$

### A.3 TRAINING SETTINGS

We run each experiment with three different splits, and report the mean and standard deviation of the performance. All experiments are run on a single V100 GPU with 16G memory. We use Adam as the optimizer and perform grid search to find the best hyper-parameter combination, with details listed in A.2. We report the baseline results directly if they are already public.

### A.4 FEATURE SPACE VISUALIZATION

To illustrate how node features form clusters in feature space, we plot the data points for 7 non-homophilous datasets, with two shown in Fig. 1. We reduce the node feature dimension to 2, sample 2000 nodes of the same class from each dataset, and plot them along the reduced dimensions. Fig. 5 shows the t-sne plots of Pokec, Genius, Cornell5, Amherst41 and US-election datasets. The plots show the clustering patterns of the nodes in feature space consistently across the most of the datasets. This observation is the main motivation for our modelling of multiple prototypes per class for transferring global information across the nodes.

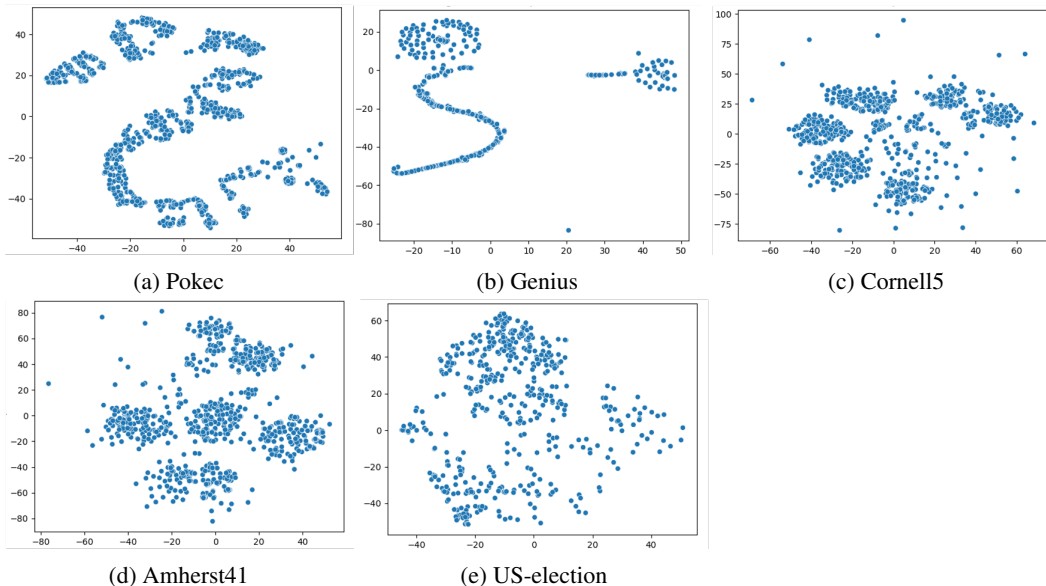

Figure 5: Feature space visualization of non-homophilous datasets.

### A.5 FURTHER ABLATION STUDY

Self attention is a classic mechanism that can be used to learn the class prototypes. In this ablation study, we discuss in detail the rationality of using slot attention instead of self attention in our framework. In principle, slot attention is preferred over self attention because of the following reasons: Firstly, In slot attention, each slot/prototype will learn to compete for the explanation via the softmax-based attention mechanism. Therefore, the learning of multiple slots/prototypes serves as soft clustering of the training nodes. Self attention, on the other hand, does not act as soft clustering. As shown in Fig. 1(a) and Fig. 6, visualized features for Penn94 and Twitch Gamers do tend to form clusters in feature space. It is beneficial to be able to capture such clustering information, as illustrated in Figure 4. Secondly, Slot attention is more efficient. In self attention, an attention score is computed for every pair of nodes, and thus the number of computations needed is $O(n^2)$. However, in slot attention, we only need to compute the attention score between nodes and slots/prototypes,

so the number of computations needed is $O(nk)$, where $k$ is the number of prototypes and is much smaller than the number of nodes $n$.

We conduct an additional ablation study on self-attention vs slot attention to empirically verify our claims. We replace slot attention with self-attention in ProtoGNN and document the performance on 7 datasets in the following table. While ProtoGNN-SA with self-attention outperforms the backbone on most of the datasets, ProtoGNN with slot attention still outperforms it on all 7 datasets.

Table 8: Classification performance comparison against baselines. Notations follow style in Tab.2.

|  | Penn94 | Pokec | Genius | Twitch-gamers | Cornell5 | Amherst41 | US-election |
|---|---|---|---|---|---|---|---|
| Backbone | 84.71 (0.52) | **82.04** (0.07) | 90.77 (0.27) | 66.06 (0.19) | 83.46 (0.61) | 81.73 (1.94) | 84.08 (0.67) |
| ProtoGNN-SA | 83.53 (0.28) | 80.25 (0.45) | 90.41 (0.24) | 66.31 (0.28) | 83.25 (0.14) | 82.15 (1.07) | 84.90 (0.65) |
| ProtoGNN | **86.23** (0.34) | 81.94 (0.10) | **91.18** (0.13) | **66.42** (0.34) | **84.14** (0.13) | **82.81** (1.34) | **86.97** (1.21) |

To verify the computation efficiency empirically, we report the average runtime of ProtoGNN with self-attention and slot attention respectively. ProtoGNN with slot attention (10.14 milliseconds) runs much faster than ProtoGNN-SA (5.76 milliseconds).

## A.6 HOMOPHILY MATRIX FOR HOMOPHILOUS DATASETS

Below are the homophily matrices for three well-known homophilous datasets: Cora, Citeseer and Pubmed (Yang et al., 2016). High homophily is signified by the high numbers in diagonal cells, whereas values of non-diagonal cells are mostly less than 0.1. This is different from the homophily matrices of non-homophilous datasets in Tab. 2, where values of non-diagonal cells are similar or even higher than diagonal cells.

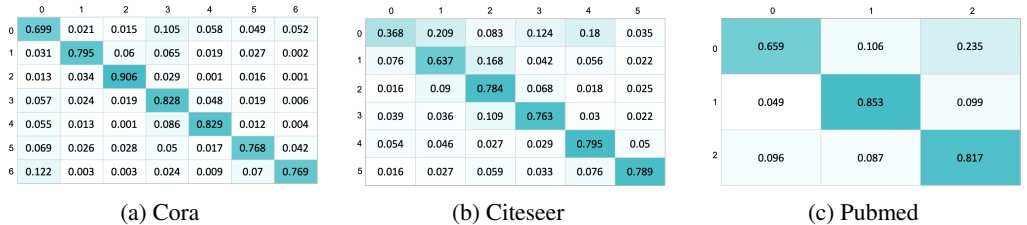

(a) Cora        (b) Citeseer        (c) Pubmed

Figure 6: Homophily matrix for three homophilous datasets.

