# OpenReview forum: "ProtoGNN: Prototype-Assisted Message Passing Framework for Non-Homophilous Graphs"
_ICLR.cc/2023/Conference — Submitted to ICLR 2023_

### Official Review · Reviewer_A8KJ · 2022-10-23

**Confidence:** 4
**Correctness:** 3
**Technical Novelty And Significance:** 2
**Empirical Novelty And Significance:** 2
**Recommendation:** 5

**Clarity, Quality, Novelty And Reproducibility:**

The writing of the paper is clear.

The overall quality is good.

Novelty is somewhat limited as
(1) slot attention is proposed by prior methods,
(2) the rationality of choosing slot attention rather than classic self-attention is not discussed.

Reproducibility looks good as the implementation details are presented.

**Strength And Weaknesses:**

Strengths:
1. Using multiple prototypes for each class is interesting.
2. The experiments are conducted on both homophilous and heterophilous graphs.
3. The writing of the paper is clear in general.


Weakness:
1. It is not quite clear how does self-attention perform. Self-attention is cheaper and more classic than slot-attention, and thus it is necessary to compare slot-attention with self-attention.
2. ProtoGNN only significantly outperforms the SOTA method GloGNN++ on 3 over 7 datasets: Penn94, Genius and US-election. It is necessary to conduct t-tests for Twitch-gamers, Cornell5 and Amherst41.

**Summary Of The Paper:**

This paper introduces a ProtoGNN for non-homophilous (or heterophilous) graphs, which could augment existing GNNs by combining node features with structural information. ProtoGNN could learn multiple prototypes for each class with slot attention. To further exploit the power of multiple prototypes, this paper uses two regularization losses: compatibility loss and orthogonality loss. The experimental results show that the proposed method is useful for several datasets.

**Summary Of The Review:**

Overall, the proposed approach is interesting and the writing is clear. However, it is not quite clear why the paper uses slot attention rather than self-attention. Besides, the experimental results in Table 4 show that ProtoGNN only significantly outperforms GloGNN++ on 3 out of 7 datasets.

---

> ### Author Response · Authors · 2022-11-13
> **Author response to Reviewer A8KJ**
>
> Thank you for your constructive comments. Please find our responses below.
>
> + **Q1. It is not quite clear how does self-attention perform. Self-attention is cheaper and more classic than slot-attention, and thus it is necessary to compare slot-attention with self-attention.**
>
>     We appreciate your feedback as we have not thought about using self-attention to learn prototype representation from training nodes.
>
>     In principle, slot attention is preferred over self attention because of the following reasons:
>
>     1. In slot attention, each slot/prototype will learn to compete for the explanation via the softmax-based attention mechanism. Therefore, the learning of multiple slots/prototypes serves as soft clustering of the training nodes. Self attention, on the other hand, does not act as soft clustering. As shown in Figure 1(a), visualized features in Penn94 and Twitch Gamers do tend to form clusters in feature space. It is beneficial to be able to capture such clustering information, as illustrated in Figure 4.
>     1. Slot attention is more efficient. In self attention, an attention score is computed for every pair of nodes, and thus the number of computations needed is $O(n^2)$. However, in slot attention, we only need to compute the attention score between nodes and slots/prototypes, so the number of computations needed is $O(nk)$, where $k$ is the number of prototypes and is much smaller than the number of nodes $n$.
>
>     Following your advice, we conduct an additional ablation study on self attention vs slot attention to empirically verify our claims. We replace slot attention with self attention in ProtoGNN and document the performance on 7 datasets in the following table. While ProtoGNN with self attention outperforms the backbone on most of the datasets, ProtoGNN with slot attention still outperforms it on all 7 datasets. We have added this to the ablation study in Section 5.2.4, and added a more detailed discussion in Appendix.
>
>
>     |                | Penn94 | Pokec | Genius | Twitch-gamers | Cornell5 | Amherst41 | US-election |
>     |----------------|--------|-------|--------|---------------|----------|-----------|-------------|
>     | backbone       | 84.71  | 82.04 | 90.24  | 66.06         | 83.42    | 81.57     | 84.08       |
>     | self attention | 83.53  | 80.25 |   90.41  | 66.31         | 83.25    | 82.15     | 84.90       |
>     | ProtoGNN       | 86.23  | 81.94 | 91.18  | 66.42         | 84.14    | 82.81     | 86.97       |
>
>     To verify the computation efficiency empirically, we report the average runtime of ProtoGNN with self-attention and slot attention respectively. ProtoGNN with slot attention runs much faster than self-attention.
>
>     | milliseconds | self-attention | slot-attention |
>     |--------------|----------------|----------------|
>     | runtime      | 10.14          | 5.76           |
>
> + **Q2. ProtoGNN only significantly outperforms the SOTA method GloGNN++ on 3 over 7 datasets: Penn94, Genius and US-election. It is necessary to conduct t-tests for Twitch-gamers, Cornell5 and Amherst41.**
>
>     Following your suggestion, we conduct t-tests with p-value=0.05. On Penn94, Genius, and US-election, we reject the null hypothesis that the two means are equal, but accept the null hypothesis on Twitch-gamers, Cornell5 and Amherst41. This suggests that ProtoGNN significantly outperforms or is comparable with the SOTA method on these datasets.
>
>     We also want to emphasize that ProtoGNN is a framework orthogonal to existing GNN models, and is therefore better evaluated based on its performance improvement when applied to these models. This is why we have conducted extensive experiments comparing vanilla GNN models against the same models boosted by ProtoGNN, with results documented in Table 2. Conducting t-tests with p-value=0.05 in Table 2, we find there are statistically significant improvements in 17 out of 23 cases when ProtoGNN boosts backbone performance. The results prove the effectiveness of ProtoGNN on various GNN backbones.

---

> ### Author Response · Authors · 2022-11-21
> **Follow-up message to Reviewer A8KJ**
>
> Thank you very much for your constructive and thoughtful review.
>
> We hope that our updated manuscript and detailed responses have addressed your comments and suggestions. It would be very helpful if you could acknowledge our response and update the scores, if satisfied. If there are any concerns remaining, we would be happy to respond.

---

### Official Review · Reviewer_mEwF · 2022-10-24

**Confidence:** 2
**Correctness:** 4
**Technical Novelty And Significance:** 1
**Empirical Novelty And Significance:** Not applicable
**Recommendation:** 5

**Clarity, Quality, Novelty And Reproducibility:**

- The clarity and reproducibility are good.
- The novelty and originality are weak.

**Strength And Weaknesses:**

Strength

- The writing and organization are good.
- The experimental evaluations are sufficient.

Weakness

- The novelty is very limited. It is not novel to introduce cluster prototypes into the transformer-based GNNs such as [1]

[1] Junjie Xu, Enyan Dai, Xiang Zhang, Suhang Wang. “HP-GMN:Graph Memory Networks for Heterophilous Graphs” Accepted by The IEEE International Conference on Data Mining (ICDM 2022)

**Summary Of The Paper:**

This paper extends the transform-based GNN by enhance the memory with multiple cluster prototypes. It justifies the correctness by effective considering the nodes from the same class but with long range. Experimental evaluations demonstrate its effectiveness.

**Summary Of The Review:**

My main concern is the limited novelty. It is not novel to introduce cluster prototypes into the transformer-based GNNs, although this may be the first which considers multiple prototypes.

---

> ### Author Response · Authors · 2022-11-13
> **Author response to Reviewer mEwF**
>
>
> Thank you for your feedback. Please find our responses below.
>
> + **The novelty is very limited. It is not novel to introduce cluster prototypes into the transformer-based GNNs such as [1]**
>
>     - Firstly, we would like to mention that the paper cited by the Reviewer, HP-GMN [1], has been only available on **Arxiv from October 15, 2022 which is well after the ICLR-2023 submission deadline. Therefore, we believe that comparing our work with HP-GMN for novelty is not fair**.
>
>     - Secondly, we found substantial differences between the two works. In terms of the introduction of cluster prototypes, HP-GMN does not claim to learn cluster prototypes as we do in the ProtoGNN framework. HP-GMN maintains a memory module which is a set of K memory vectors that can capture global patterns in the graph. These memory vectors are substantially different from cluster prototypes both in motivation and functionility, as discussed below.
>
>       1. Memory units in HP-GMN are intended to represent global patterns. However, there is no inductive bias to make sure that the global patterns represent node clusters. In fact, the word "cluster" does not even appear in the paper. Therefore, the clustering motivation of ProtoGNN seems orthogonal to the motivation of memory units in HP-GNN.
>       1. The memory units in HP-GMN are trained to learn embeddings that are representative and diverse. However, it is not clear what the memory vectors learn since there is no inductive bias in learning the memory vectors. In contrast, our method enforces the prototype representations to learn multiple clusters representing the classes with the slot-attention mechanism. Note that the slot-attention is **designed to capture clusters** in the input nodes and thus provides the needed inductive bias.
>       1. In HP-GMN, the memory units are not class-specific. It learns K memory units from all the training nodes. However, in ProtoGNN, we learn separate prototype representations for separate class labels of training nodes and are trained to represent separate classes.
>       1.  The number of memory units used *i.e.* K in HP-GMN is among {20, 50, 100, 200, 300, 500} which is very high compared to our use of the number of prototypes within {1, 3, 5, 10}. The effectiveness of a relatively small number of prototypes can be explained by the inductive bias we introduced in learning multiple/clustered class-specific prototypes as well as feature view - structure view delinking of learning which is missing in HP-GMN.
>
>     - Furthermore, we would like to reiterate that our main contribution is the framework where node features and structure are disentangled from each other for learning and multiple class-specific cluster prototypes are used to transfer long-distance information.
>
>     - In addition, our work is not built on Transformers-based GNNs. It is a generic framework that can be added to existing GNNs regardless of their architectures. This whole framework helps in learning richer node representations as shown in our several ablation experiments.
>
>     - Lastly, to improve the clarity, we have added a small discussion regarding HP-GMN in the Related work section.
>
>
>
>
>     [1] Junjie Xu, Enyan Dai, Xiang Zhang, Suhang Wang. “HP-GMN:Graph Memory Networks for Heterophilous Graphs”

---

> ### Author Response · Authors · 2022-11-21
> **Follow-up message to Reviewer mEwF**
>
> Thank you very much for your time reviewing our work.
>
> We hope that our updated manuscript and detailed response have addressed your comments regarding novelty in comparison with HP-GMN. It would be very helpful if you could acknowledge our response and update the score, if satisfied. If there are any concerns remaining, we would be happy to respond.

---

### Official Review · Reviewer_Fqiu · 2022-10-25

**Confidence:** 3
**Correctness:** 3
**Technical Novelty And Significance:** 2
**Empirical Novelty And Significance:** 2
**Recommendation:** 6

**Clarity, Quality, Novelty And Reproducibility:**

Very good clarity

Quality is OK in terms of model design, and is good in terms of experiments

Novelty is OK

Not sure about the reproducibility, but since the module is reasonable, should be reproducible

**Strength And Weaknesses:**

Strengths:

1. The paper is well written and easy to follow. The explanation on the model is very clear with a good logic.

2. The studied problem of heterophilous graph learning is meaningful.

3. The experiments are comprehensive, 10 datasets are used and some of them are very large. Also, many baselines are included in the commparison.


weakness:

1. when generating the structural view, if the GNN are the ones for homophilous graphs, then the obtained embedding should not be of high quality, and accordingly there is no reason to believe they can be used to properly aggregate the prototypes. But if the adopted GNNs are the ones for heterophilous graphs, then the mechanism proposed in this paper is only an incremental part of the whole model instead of a key module.

Therefore, whether the proposed module works well highly depends on whether the adopted GNN backbone is good enough, making the contribution of this work not very significant. Also, according to the experiments, the improvement gained by including ProtoGNN is indeed not significant.

2. what is the backbone GNN used in Table 4

3. It would be better to also show the homophily matrix of the homophilous graphs. In the current version, althoug the homophily matrix of the heterophilous graphs are given, but it seems that most of the classes have half homophilous edges and half heterophilous edges, and not sure how to understand these values. E.g. is 0.5,0.5 enough to say this class is heterophilous enough?

4. Are all the heterophilous datasets binary classification datasets with only two classes? Why is this? Isn't there any datasets with more classes?

**Summary Of The Paper:**

This paper targets the incapability of GNNs on heterophilous graphs and proposes to process the node feature and graph structures separately. Specifically, Multiple node feature prototypes are first generated with only the node feature information. Then, arbitrary GNN is used to obtain a structural view (embedding) of each node. Finally, the prototypes are aggregated into the structural embeddings via an attention based mechanism. Experiments are conducted on multiple datasets and compared with multiple baselines.


**Summary Of The Review:**

Overall, this paper is well written and targets a meaningful problem, but the contribution is not significant, because the performance is mainly determined by the backbone, which is also shown in experiments.

Therefore, I think this paper is only slightly above acceptance threshold

---

> ### Author Response · Authors · 2022-11-13
> **Author response to Reviewer Fqiu (Part 1 of 2)**
>
>
> Thank you very much for your insightful and valuable comments. Please find below our responses for specific points.
>
> + **Q1. When generating the structural view, if the GNN are the ones for homophilous graphs, then the obtained embedding should not be of high quality, and accordingly there is no reason to believe they can be used to properly aggregate the prototypes.**
>
>     We respectfully disagree that the obtained embeddings are not of high quality if the underlying GNN backbones are for homophilous graphs. With the assistance of additional information transfer from the prototypes, our framework influences the way backbone GNNs learn. In the structure view, the representations generated after each convolutional layer are adapted to match with the prototypes from the feature-view allowing it to adapt better to label heterogeneity. With this, the prototype nodes provide the signal to the GNN backbone in the structure view and modulate their learning which is likely to make the backbone GNN ignore the not-useful local information. This is noticeable in the experiments, as shown in Table 2, where regardless of the underlying GNN, we consistently improve the results. Specifically, GCNII+ProtoGNN achieves new SOTA on Genius dataset.
>
>     We notice that Figure 2 from the original paper does not clearly explain this fact since it draws the structure view learning and information fusion separately and may cause confusion that the two components take place sequentially. However, as stated in our original writeup on Page 5 (after Equation 6), the information transfer from prototypes can happen after any layer of the structure view. We have modified Figure 2 along with its captions to better reflect the actual working of the method. In fact, in the implementation of ProtoGNN on the GCN backbone, we use a 2-layer GCN and conduct information fusion in between these 2 layers.
>
>
>
> + **Q2.But if the adopted GNNs are the ones for heterophilous graphs, then the mechanism proposed in this paper is only an incremental part of the whole model instead of a key module.**
>
>     We would like to reiterate that our framework is designed to work with any backbone GNN agnostic to whether it works well on heterophilous graphs or not. Our experiments demonstrate that we consistently and substantially improve upon both types of GNNs (such as GCN and LinkX) as shown in Table 2. Specifically, the performance improvement ProtoGNN brings to GNNs designed for heterophilous graphs is not trivial. We have conducted t-tests (p-value=0.05) to compare the performance between LINKX and LINKX+ProtoGNN and found the improvement statistically significant in 4 out of 6 cases.
>
>
> + **Q3. What is the backbone GNN used in Table 4**
>
>     As specified in Section 5.2.2, All ProtoGNN results in the original Tab. 4 (now Tab. 3) are with LinkX backbone except for Genius where GCNII is the backbone.

---

> > ### Author Response · Authors · 2022-11-13
> > **Author response to Reviewer Fqiu (Part 2 of 2)**
> >
> > + **Q4. It would be better to also show the homophily matrix of the homophilous graphs. In the current version, although the homophily matrix of the heterophilous graphs are given, but it seems that most of the classes have half homophilous edges and half heterophilous edges, and not sure how to understand these values. E.g. is 0.5,0.5 enough to say this class is heterophilous enough?**
> >
> >     Thanks for your feedback. To improve the clarity of the paper, we plot the homophily matrices for three well-known homophilous datasets, Cora, Citeseer, and Pubmed, and add a section in the Appendix.
> >
> >     As illustrated in the homophily matrices below, and as discussed in LinkX paper, the homophilous graphs have dominant diagonal values in the homophily matrix, a 0.5, 0.5 would be considered as heterophilous.
> >
> >     Cora
> >     |   | 0     | 1     | 2     | 3     | 4     | 5     | 6     |
> >     |---|-------|-------|-------|-------|-------|-------|-------|
> >     | 0 | **0.699** | 0.021 | 0.015 | 0.105 | 0.058 | 0.049 | 0.052 |
> >     | 1 | 0.031 | **0.795** | 0.06  | 0.065 | 0.019 | 0.027 | 0.002 |
> >     | 2 | 0.013 | 0.034 | **0.906** | 0.029 | 0.001 | 0.016 | 0.001 |
> >     | 3 | 0.057 | 0.024 | 0.019 | **0.828** | 0.048 | 0.019 | 0.006 |
> >     | 4 | 0.055 | 0.013 | 0.001 | 0.086 | **0.829** | 0.012 | 0.004 |
> >     | 5 | 0.069 | 0.026 | 0.028 | 0.05  | 0.017 | **0.768** | 0.042 |
> >     | 6 | 0.122 | 0.003 | 0.003 | 0.024 | 0.009 | 0.07  | **0.769** |
> >
> >     Citeseer
> >     |   | 0     | 1     | 2     | 3     | 4     | 5     |
> >     |---|-------|-------|-------|-------|-------|-------|
> >     | 0 | **0.368** | 0.209 | 0.083 | 0.124 | 0.18  | 0.035 |
> >     | 1 | 0.076 | **0.637** | 0.168 | 0.042 | 0.056 | 0.022 |
> >     | 2 | 0.016 | 0.09  | **0.784** | 0.068 | 0.018 | 0.025 |
> >     | 3 | 0.039 | 0.036 | 0.109 | **0.763** | 0.03  | 0.022 |
> >     | 4 | 0.054 | 0.046 | 0.027 | 0.029 | **0.795** | 0.05  |
> >     | 5 | 0.016 | 0.027 | 0.059 | 0.033 | 0.076 | **0.789** |
> >
> >     PubMed
> >
> >     |   | 0     | 1     | 2     |
> >     |---|-------|-------|-------|
> >     | 0 | **0.659** | 0.106 | 0.235 |
> >     | 1 | 0.049 | **0.853** | 0.099 |
> >     | 2 | 0.096 | 0.087 | **0.817** |
> >
> >
> > + **Q5. Are all the heterophilous datasets binary classification datasets with only two classes? Why is this? Isn't there any datasets with more classes?**
> >
> >     We chose to work on large heterophilous datasets proposed in recent works. These datasets used to have multiple classes, but previous authors changed the labels to binary in order to create datasets with label heterogeneity. We believe the number of classes would not affect the effectiveness of our work.

---

> > > ### Comment · Reviewer_Fqiu · 2022-11-19
> > > **post-rebuttal response**
> > >
> > > Thanks for the responses and the additional information. I'm overall sstisfied and think this paper is above acceptance threshold, so I will keep my rating

---

> > > > ### Author Response · Authors · 2022-11-19
> > > > **Author reply**
> > > >
> > > > Thank you very much for your positive response to our work! We appreciate your thoughtful comments which helped us improve the paper.

---

### Official Review · Reviewer_v2yj · 2022-10-25

**Confidence:** 4
**Clarity, Quality, Novelty And Reproducibility:** The paper is not difficult to follow,…
**Correctness:** 2
**Technical Novelty And Significance:** 3
**Empirical Novelty And Significance:** 2
**Recommendation:** 5

**Strength And Weaknesses:**

Node label heterogeneity raise unique challenge for effectively learning node representation. This work takes an important step to deal with it by not fully relying on a pure GNN model. However, there are still important questions that are not answered.

Strength:
1. A prototype-based method is proposed to deal with graph heterogeneity issue.
2. Special regularizations are designed along with the proposed method for effectively controlling the prototype distribution.
3. Extensive experiments are conducted on multiple datasets to show the classification performance of the proposed method.

Weakness:
My main concern is about the novelty and motivation behind this work. More specifically, it's difficult for me to clearly catch up with the motivation and exact contribution to advancing GNN method for dealing with label heterogeneity issue. More specially, I have following concerns.

1. The justification in the Introduction section is too weak to explaining why prototype can deal with the limitations mentioned in the second and third paragraphs.

 2. It deserves more words to explain why we need multiple prototypes for each classes, and what is the connection between aggregating features from distant nodes and solving non-homophilous problem. The difference from previous works which also uses prototype or clustering centers is not discussed.

3. Besides, it's difficult to understand what is the contribution of this work to advancing GNNs for better adapting to non-homophilous dataset. From the architecture shown in this work, we can see that the key component is a MLP-based encoder, while GNN layer can be initialized any GNN model. What if the selected GNN module like vanilla GCN, GAT etc. can not deal with the label heterogeneity problem?


**Summary Of The Paper:**

This work attempts to deal with node heterogeneity for graph learning task. The proposed method dissects the node representations into feature and structure views. Finally a prototype-based node representation method is designed by separately modeling feature view with prototype-based MLP network and encoding graph structure with GNN.

**Summary Of The Review:**

Though experimental results show promising aspects of the proposed method, it still needs a significant improvement over the presentation quality, especially the clarity of motivation.

---

> ### Author Response · Authors · 2022-11-13
> **Author response to Reviewer v2yj (Part 1 of 2)**
>
> Thank you very much for your insightful and constructive comments. Please find below our responses for specific points.
>
>
> +    **Q1. The justification in the Introduction section is too weak to explaining why prototype can deal with the limitations mentioned in the second and third paragraphs.**
>
>         Our framework addresses the two limitations by disentangling feature and structure views and adaptively learning the embeddings from both views. Below we state the two limitations of GNNs on heterophilous graphs we discussed and how ProtoGNN addresses the limitations.
>         + Insuffificent exploitation of node features:
>
>             In the feature view, ProtoGNN captures the node feature information into multiple prototypes per class **without dilution from GNN message passing**.
>         + Local useful information is scarce and we need to capture global information efficiently:
>
>             In the structure view, the GNNs adaptively learn the node embeddings from the local neighborhood as well as from the prototype nodes. The message passing from the prototypes is equivalent to introducing artificial edges from training nodes of each class to all non-training nodes via prototype nodes. This serves as a shortcut in message passing which bypasses local graph neighborhoods and captures global information. The shortcut enables ProtoGNN to leverage the strong node feature information even from distant nodes beyond local neighborhoods. Note that since this interaction between feature view and structure view can happen at each iteration of GNN message passing, this enables the GNNs to adaptively capture local as well as global information.
>
>         We thank the reviewer for pointing out that we did not do a good job explaining this in the introduction. To fix this, we have added an additional paragraph explaining how our framework directly addresses the limitations:
>
>         The learning of prototypes introduces artificial edges from training nodes of each class to all non-training nodes via prototype nodes. This serves as a shortcut in message passing which bypasses local graph neighborhoods and captures global information, while maintaining the linear complexity with respect to the backbone GNNs. Additionally, since the prototypes are learned from node feature space, it preserves strong feature information that might be diluted by traditional message passing.
>
>
> +    **Q2. It deserves more words to explain why we need multiple prototypes for each classes**
>
>         Thanks for your constructive suggestion. We have modified the introduction accordingly and added the following paragraph:
>
>         We observe that in many datasets, nodes from the same class form multiple clusters. If we use a single prototype for each class, the prototype representation would be inaccurate or insufficient. For example, the single prototype may either learn to represent the average of different clusters. Alternatively, the single prototype may learn to represent one of the dominant clusters in the feature space, therefore failing to capture information about the remaining clusters. Hence using a single prototype for each class may lead to suboptimal performance.
>
>         We confirm this hypothesis in the ablation experiment of section 5.2.4, where we vary the number of prototypes and observe a visible drop in performance when there is only one prototype per class.
>
>
> +    **Q3. What is the connection between aggregating features from distant nodes and solving non-homophilous problem.**
>
>         In non-homophilous graphs, a node's features tend to be dissimilar to its neighbors'. Thus, when the node representation is learned by convoluting its neighbors whose features are dissimilar, the node's features get diluted.
>
>         This can be mitigated if the GNNs are fed a signal to align the node representation with the clustered representations of the nodes of the same class. By attending to prototypes from the same class during information fusion, the node representation can adaptively learn from the statistics of the whole class, which are represented by the prototypes as clustered information. This will make the final node representation more aligned with the representations of the same class and therefore more discriminative.
>
>
> +    **Q4. The difference from previous works which also uses prototype or clustering centers is not discussed.**
>
>         Thanks for your suggestion. Following your advice, we have modified the related work section to incorporate the discussion of these works.

---

> > ### Author Response · Authors · 2022-11-13
> > **Author response to Reviewer v2yj (Part 2 of 2)**
> >
> >
> > +    **Q5. Besides, it's difficult to understand what is the contribution of this work to advancing GNNs for better adapting to non-homophilous dataset. From the architecture shown in this work, we can see that the key component is a MLP-based encoder, while GNN layer can be initialized any GNN model.**
> >
> >
> >         - Our key contribution is the full framework that disentangles the feature view and structure view to enable more adaptive learning on non-homophilous graphs. The structure view, information fusion, and node multi-prototype learning in the feature view are all key components of the framework, aside from the MLP-based encoder.
> >
> >
> >         - In the structure view, while each GNN layer can be initialized by any GNN model, the structure view node representation learning is still different from the original GNN backbone. This is because information fusion with the feature view can happen at any layer of the GNN model. For example, in the case of GCN+ProtoGNN from Table 2, information fusion takes place between the two convolutional layers. Integrating information fusion into the structure view node representation learning allows the representation to be learned with the assistance of the feature view information, therefore avoiding the message dilution and the trap of the local neighborhood.
> >
> >
> >         - The fact that each GNN layer can be initialized by any GNN model is a feature of our framework. We are motivated to design a generic framework that can be added to existing GNN models to boost their performance on non-homophilous graphs, regardless of their architectures. As explained in section 4, our framework  preserves the efficiency of the original GNN backbones by adding only linear complexity.
> >
> >
> >         - We understand that we may have caused the misunderstanding that structure view learning and information fusion take place sequentially because the two components were drawn separately in Figure 2. We have modified Figure 2 along with its captions to better reflect the actual working of the method, in congruence with the paragraph below Equation 6 on Page 5 of the original paper.
> >
> >
> >         - Node multi-prototype learning is also key to our framework. We adapt slot attention to enable multiple prototypes per class to compete for an explanation of each training node. Previous works on prototype learning on graphs do not learn multiple prototypes per class. The multiple prototypes serve as soft clustering and are able to capture multi-modal distribution in feature space. We explain why multiple prototypes are needed in the answer to Q2.
> >
> >
> >         - All these components are learned end-to-end. The combined framework is intended as a generic component that can bring consistent performance boosts to past and future GNN models.
> >
> >
> >
> > +    **Q6. What if the selected GNN module like vanilla GCN, GAT etc. can not deal with the label heterogeneity problem?**
> >
> >
> >         - ProtoGNN is orthogonal to all existing GNN models, be it models for homophilous or non-homophilous graphs. It can be added to existing GNN models to boost their performance on non-homophilous graphs regardless of their model architectures.
> >
> >
> >         - With additional information transfer from the prototypes, our framework influences the way backbone GNNs learn. In the structure view, the representations generated after each convolutional layer are adapted to match with the prototypes from the feature view, allowing it to adapt better to label heterogeneity. With this, the prototype nodes provide the signal to the GNN backbone in the structure view and modulate their learning which is likely to make the backbone GNN ignore the not-useful local information. This is  noticeable in the experiments, as shown in Table 2, where regardless of the underlying GNN, we consistently improve the results.

---

> ### Author Response · Authors · 2022-11-21
> **Follow-up message to Reviewer v2yj**
>
> Thank you very much for your constructive and thoughtful review.
>
> We hope that our updated manuscript and detailed responses have addressed your comments and suggestions. It would be very helpful if you could acknowledge our response and update the scores, if satisfied. If there are any concerns remaining, we would be happy to respond.

---

### Author Response · Authors · 2022-11-13
**General Response**

We thank all the reviewers for their time and constructive feedback! We have updated the manuscript addressing the comments of the reviewers and have uploaded the revised version. In summary, we have improved our paper with the following:

1. Page 2: Added a paragraph explaining of why using a single prototype per class is insufficient.
1. Page 2: Added a paragraph discussing how ProtoGNN addresses the two limitations identified.
1. Page 3: Added a discussion of the differences from previous works which use prototype-based learning in graphs in Related Work.
1. Page 4: Modified Figure 2 along with its captions to better reflect the actual working of our method, in congruence with the paragraph below Equation 6 on Page 5 of the original manuscript.
1. Page 7: Merged Table 2 and Table 3 of the original manuscript into Table 2 to save space.
1. Page 7: Table 4 of the original manuscript now becomes Table 3 due to the merging of tables.
1. Page 9: Added an ablation study of using self attention vs slot attention.
1. Appendix: Added a detailed discussion of ablation on self attention vs slot attention.
1. Appendix: Added the homophily matrices of three well-known homophilous datasets with further explanations of the metric.

In order to make space, we have deleted some sentences and adjusted the size of tables and figures.

We hope our modifications and clarifications have addressed most of the concerns about our paper. We would be happy to discuss and address any further concerns you may have.

---

### Author Response · Authors · 2022-11-17
**Thank you**

We thank the reviewers again for their time providing constructive feedback on our work.

We hope that our responses have addressed most of the concerns raised by the reviewers. We would be happy to discuss any additional questions or concerns you may have. As there are less than two days until the discussion period ends, it would be much appreciated if you could soon share with us any further questions for us to respond within time.

---

### Decision · Program_Chairs · 2023-01-20

**Decision:**

Reject

**Justification For Why Not Higher Score:**

*  The reviewers agree that novelty is somewhat limited as slot attention is proposed by prior methods, and the rationality of choosing slot attention rather than classic self-attention is not discussed.
* Further consensus exists on the lack of motivation behind this work. It is unclear why multiple prototypes for each class are needed and the connection between aggregating features from distant nodes and solving a non-homophilous problem. The difference from previous studies, which also use prototype or clustering centers, is not discussed.

**Justification For Why Not Lower Score:**

N/A

**Metareview: Summary, Strengths And Weaknesses:**

Node label heterogeneity raises a unique challenge for effectively learning node representation. This work takes an important step to deal with it by not fully relying on a pure GNN model. It described an approach that divides node representations into feature and structure views and designs prototype-based node representations by separately modeling feature views with prototype-based MLP and graph structure with GNN.

**Strengths:**
* The overall quality is good. The writing of the paper is clear.
* Reproducibility looks good as the implementation details are presented.
* Experiments are comprehensive, and some of them are very large. However, reviewers note that all heterophilic datasets are binary classification datasets with only two classes.

**Weaknesses:**
*  The reviewers agree that novelty is somewhat limited as slot attention is proposed by prior methods, and the rationality of choosing slot attention rather than classic self-attention is not discussed.
* Further consensus exists on the lack of motivation behind this work. It is unclear why multiple prototypes for each class are needed and the connection between aggregating features from distant nodes and solving a non-homophilous problem. The difference from previous studies, which also use prototype or clustering centers, is not discussed.

I'd like to note that one reviewer's score driving concern was the high similarity between this paper and another paper that was only made available on Arxiv on 15 Oct 2022, well after the ICLR-2023 submission deadline in late September. However, even excluding that review from consideration, reviewers are not enthusiastic about this submission.